# You Are What You Eat—The Relationship between Diet, Microbiota, and Metabolic Disorders—A Review

**DOI:** 10.3390/nu12041096

**Published:** 2020-04-15

**Authors:** Małgorzata Moszak, Monika Szulińska, Paweł Bogdański

**Affiliations:** Department of Treatment of Obesity, Metabolic Disorders and Clinical Dietetics, Poznan University of Medical Sciences, 61-569 Poznań, Poland; mszulinska1@wp.pl (M.S.); pbogdanski@ump.edu.pl (P.B.)

**Keywords:** gut microbiota, diet, metabolic disorders, obesity, dyslipidemia, diabetes, short-chain fatty acid, vegetarian, Western-style diet

## Abstract

The gut microbiota (GM) is defined as the community of microorganisms (bacteria, archaea, fungi, viruses) colonizing the gastrointestinal tract. GM regulates various metabolic pathways in the host, including those involved in energy homeostasis, glucose and lipid metabolism, and bile acid metabolism. The relationship between alterations in intestinal microbiota and diseases associated with civilization is well documented. GM dysbiosis is involved in the pathogenesis of diverse diseases, such as metabolic syndrome, cardiovascular diseases, celiac disease, inflammatory bowel disease, and neurological disorders. Multiple factors modulate the composition of the microbiota and how it physically functions, but one of the major factors triggering GM establishment is diet. In this paper, we reviewed the current knowledge about the relationship between nutrition, gut microbiota, and host metabolic status. We described how macronutrients (proteins, carbohydrates, fat) and different dietary patterns (e.g., Western-style diet, vegetarian diet, Mediterranean diet) interact with the composition and activity of GM, and how gut bacterial dysbiosis has an influence on metabolic disorders, such as obesity, type 2 diabetes, and hyperlipidemia.

## 1. Introduction

In recent years, there has been a growing interest in the role of gut microbiota (GM) and their implications for human health, especially in the context of metabolic disorders (obesity, type 2 diabetes, dyslipidemia), which increase the risk of cardiovascular incidents and thus the number of deaths in developed countries. One of the most important factors affecting the state of the microbiota is undoubtedly diet. This manuscript discusses the impact of different types of diets on the microbiota and the relationship between diet, the state of the intestinal bacterial flora, and the development of selected metabolic disorders.

The gut microbiota is defined as the community of microorganisms (bacteria, archaea, fungi, viruses) colonizing the gastrointestinal tract [1,2]. The number and type of microorganisms differ along the gastrointestinal tract, with their distribution being determined by pH, oxygen and nutrient availability, digestive flow rates, and secreted enzymes. For example, the concentration of bacteria in the stomach is relatively small (10 CFU/g), increasing steadily to 10^7^ CFU/g in the ileum and 10^12^ CFU/g in the colon [3]. The upper section of the digestive tract is colonized mainly by aerobic bacteria, while the lower section is mainly populated by anaerobic bacteria. The main clusters residing in the human digestive tract are from five phyla—Firmicutes (79.4%; *Ruminococcus, Clostridium, Eubacteria*) and Bacterioidetes (16.9%; *Porphyromonas, Prevotella*), followed by phyla Proteobacteria (1%), Actinobacteria (2.5%; *Bifidobacterium*), and Verrumicrobia (0.1%) [4,5]. In the proximal gut, the dominant groups are *Lactobacillus* (Firmicutes), *Veillonella* (Firmicutes), and *Helicobacter* (Proteobacteria). In the duodenum, jejunum, and ileum, the most numerous groups are *Bacilli* (Firmicutes), *Streptococcaceae* (Firmicutes), and *Actinomycinaeae*, and in the colon, an increased proportion of *Lachnospiraceae* (Firmicutes) and *Bacteroidetes* are observed [6]. GM regulates various metabolic pathways in the host, including those involved in energy homeostasis, glucose metabolism, and lipid metabolism [6]. Because of its broad metabolic activity, GM is often called “a new virtual metabolic organ” [7]. Previous studies have shown that GM plays important roles in nutrient degradation and adsorption [3], short-chain fatty acids (SCFAs), amines, phenols/indoles, and sulfurous compounds production [8], vitamin B and K synthesis [9], the bioavailability of minerals, and the metabolism of bile acids [10]. GM helps maintain gut integrity by stabilizing cell-cell junctions, and it acts in restoring the epithelial barrier after damage [11]. Maintaining a proper microbiota composition plays a critical role in protection against pathogens and is an integral part of the overall host immune response [6]. By regulating the release of neurotransmitters and other neuroactive substances (serotonin, dopamine, γ-aminobutyric acid (GABA)), GM influences the central nervous system [12].

Multiple factors modulate the composition of the microbiota and its activity. In humans, the gut flora evolves in several stages, but the most dynamic period in GM establishment is the first one to two years of life. The microbial pattern shaping in the first 2–5 years of life will then determine the GM profile in later stages of development—in early childhood when GM grows and diversifies, and in adolescence when the community of bacteria in the gastrointestinal tract stabilizes [13]. GM profiles vary between different races/ethnicity and sex/gender [14]. The individual microbiota pattern is influenced by antibiotic use (especially in the first years after birth) [15,16], medication (nonsteroidal anti-inflammatory drugs, proton pump inhibitors), infections, and chronic stress. The size, species composition, and diversity of bacteria in the human digestive tract are also shaped by host genotype, physical activity level, personal hygiene, and xenobiotics [17]. But one of the most significant roles is played by nutrition: composition of the diet, dietary pattern, and long-term dietary habits (consumption of snacks and junk food, late-night eating, breakfast skipping, nutritional habits) [18,19]. 

In this paper, we reviewed the current knowledge about the relationship between nutrition, gut microbiota, and host metabolic status. We described how diet interacts with the composition and physiological activity of GM and how gut dysbiosis influences metabolic disorders, such as obesity, type 2 diabetes, and hyperlipidemia.

## 2. Diet and Gut Microbiota

Diet affects multiple aspects of human health. It is well documented that improper nutrition patterns, e.g., a Western-style diet (WSD) or a high-fat diet (HFD), are linked to chronic diseases of civilization, such as obesity, type 2 diabetes, and cardiovascular disease (CVD) [19]. Long-term nutritional habits are essential not only for determining the human health status but also for maintaining high diversity and abundance of microbial populations in the GI tract, termed “eubiosis” [20]. 

### 2.1. Infant’s Diet and Gut Microbiota Establishment

The first 2–5 years of life play a significant role in determining the GM. An infant’s microbiome will closely resemble their mother’s microbiome, which results from influences by many of the maternal microbiomes—mouth, skin, vagina, gastrointestinal (GI) tract, and breastmilk [21]. One of the most crucial factors contributing to GM in childhood is diet. Several studies have discovered that distinct infant microbiome profiles correlate with different nutritional aspects, such as breastfeeding, formula-feeding (formula ingredients) [22], maternal gestational diet [22], time of introduction of solid foods, or model of diet weaning [23,24]. 

World Health Organization recommends breastfeeding as the best method for infant nutrition for the first six months of life and then continued supplemental breastfeeding up to 2 years and beyond [22]. Breastfeeding plays an essential role in infant metabolic and immunological programming and positively influences the microbiota diversity and composition. The variety of the gut microbial community in the first month of life is rather small, but most studies indicate differences in the microbiota of formula-fed versus breast-fed infants. Newborns fed with mother’s breast milk have a higher abundance of beneficial bacteria species compared to formula-fed children. Breastfeeding is associated with favorable alteration of infant GM (high colonization with *Bifidobacterium* and *Lactobacillus*, *Staphylococcus* and *Bacteroides*, low colonization with *Clostridium, Escherichia*), in addition to inhibiting colonization by potential pathogens (*Escherichia coli*, *Helicobacter jejuni*, *Shigella, Vibrio cholera, Salmonella*), as described in several previous studies [25,26,27]. In comparison, formula-fed infants are enriched in functions typical of the adult microbiota and have higher counts of *Clostridia* (*C. paraputrificum*, *C. perfringens*, *C. clostridiiforme*, *C. difficile*, and *C. tertium*), *Streptococcus* (*S. bovis*, *S. faecalis*, and *S. faecium*), and *Proteobacteria* (*Enterobacter cloacae*, *Citrobacter spp.*, and *Bilophila wadsworthia*) than do breast-fed infants, and with delayed colonization by *Bifidobacteria* [22]. The beneficial effect of mother’s breast milk on the microbiota formation results from the content of bioactive compounds, mainly human milk oligosaccharide (HMO), HMO-related metabolites, immunological components (secretory IgA), and fatty acids [26]. Because of the “bifidogenic effect”, HMO provides optimal growth factors for promoting and supporting specific microbial establishment. The higher presence of *Bifidobacterium* spp. in early life entails protection against obesity [28]. Fatty acids and monoglycerides arising from the hydrolysis of milk triglycerides support the infant’s innate immunity against several pathogens, including *Giardia lamblia, Haemophilus influenzae, Group B streptococci, Staphylococcus epidermidis*, respiratory syncytial virus, and herpes simplex virus type 1 [29]. The positive role of breastfeeding is likely to be caused by the influence of age-appropriate and environment-adjusted gut colonization and provides numerous health benefits, such as protection against obesity, diabetes, and other components of metabolic syndrome in adulthood [30,31]. However, it should be remembered that the composition of both macronutrients and microbiota of breast milk depends on many factors, such as mother’s health [30], gestational weight gain, mode of delivery [31], duration of breastfeeding, and model of breastfeeding (exclusively breastfeeding vs. non-exclusively breastfeeding), and these are what ultimately affect the infant’s GM pattern [26].

Following breastfeeding or formula-feeding, weaning and the introduction of solid foods are associated with rapid development in the structural and functional diversity of the infant microbe community, establishing a mature, adult-like state of GM. The induction of complementary feeding and the gradual transition to the “family” diet is characterized by an increase of adult-associated microbes from the family *Lachnospiraceae* and *Ruminococcaceae* [24]. Maturation and differentiation are essential steps in GM establishment, which is why many researchers ask the question: When is the optimal time for reduction of breastfeeding and introduction of “adult” foods? 

Laursen et al. [32] showed that besides the duration of breastfeeding, the composition of the complementary diet was also a significant determinant of GM development. The introduction of family foods containing high protein and fiber strongly influenced both microbial composition and GM alpha diversity. Intake of meats, cheeses, and Danish rye bread, rich in protein and fiber, were associated with increased alpha diversity [32]. Additionally, Matsuyama et al. [33] observed significant changes in the gut microbiota community, correlated with changes in the children’s dietary pattern over a period of 12 months. They noticed differential effects on specific Firmicutes-affiliated lineages in response to intake frequency of processed or unprocessed foods and positive influence of intake of fortified milk supplemented with Bifidobacterium probiotics and prebiotics (synbiotics) on Bifidobacterium spp. [33]. 

Interestingly, an infant’s GM may also be determined by maternal gestational diet through a vertical transfer of maternal microbes to infants during vaginal delivery and breastfeeding. This theory is confirmed by the study of Lundgren et al. [34], who identified distinctions in infant microbial community structure dependent on maternal fruit and milk intake during pregnancy, but this effect differs according to the delivery model. Additionally, reports are suggesting that transmission of commensal bacteria from mother to fetus is likely to occur prior to parturition [35]. Chu et al. [35] observed that independent of maternal body mass index, a maternal high-fat diet led to distinct changes in the neonatal gut microbiome at birth, and which persisted through 4–6 weeks of age. 

The relationships between the maternal gestational diet, breastfeeding, formula feeding, weaning, and GM shaping underscore the importance of educating pregnant mothers on proper nutrition during pregnancy and lactation.

### 2.2. Diet Composition and Gut Microbiota 

Different dietary patterns, in particular, the composition of macronutrients and micronutrients of the diet and nutritional sources of macronutrients, contribute to GM remodeling. Even short-term dietary changes (a few days) may modulate GM composition and actively affect the host metabolism [36]. 

#### 2.2.1. Carbohydrates

Among the macronutrients, carbohydrates (CHOs) play the most crucial role in shaping the GM, and their effects on the bacterial community have been the best described. It has been noted that simple CHOs (e.g., sucrose, fructose) cause rapid deregulation in the composition of the GM and hence metabolic dysfunction in the host [20], while complex CHOs, specifically, certain “microbiota accessible carbohydrates” (MACs), are beneficial. Oligosaccharides, such as fructooligosaccharides (FOS) and galactooligosaccharides (GOS), have been termed prebiotic, and they represent complex CHOs with the most influence on the GM composition. Their role in promoting Bifidobacterium and Lactobacillus growth has been widely described [37,38]. The systematic review and meta-analysis (64 studies involving 2009 individuals) concerning the effect of dietary fiber interventions on gut microbiota composition in healthy adults conducted by So et al. [39] showed that dietary fiber intervention resulted in a higher abundance of *Bifidobacterium* spp. and *Lactobacillus* spp., as well as elevated fecal butyrate production compared with the low-fiber counter group. Additionally, both FOS and GOS significantly increased the count of both *Bifidobacterium* spp. and *Lactobacillus* spp. compared with the control, but did not affect microbial alpha diversity and abundances of other prespecified bacteria. Data from selected studies regarding the influence of digestible and non-digestible carbohydrates on GM establishment and functionality are shown in Table 1.

Fruits, vegetables, and wholegrain cereals are the main sources of complex carbohydrates in the human diet [51]. However, plant cell wall polysaccharides and resistant starch (RS) are difficult to digest and absorb in the intestinal tract. Therefore, they undergo microbial breakdown and subsequent fermentation to the final products: SCFAs. Butyrate is mainly produced by Firmicutes, propionate is produced by Bacteroidetes, and acetate is made by most gut anaerobes [52]. Butyrate is the primary energy source for colonocytes and plays an important role in brain function and maintenance of the intestinal barrier. Propionate is a substrate for gluconeogenesis in the liver and, along with butyrate, promotes intestinal gluconeogenesis. Acetate and butyrate are substrates for fatty acid synthesis [53]. The SCFAs have multiple effects, including inhibition of histone deacetylase activity (HDAC) (butyrate), modulation of expression of peroxisome proliferator-activated γ receptor (PPAR-γ), G-protein-coupled receptor 43 (GPR43; or free fatty acid receptor 2 (FFA2)) and GPR41 (or FFA3) (all three SCFAs), and of GPR109A (only butyrate). Acetate impacts intestinal antibody IgA responses, and butyrate inhibits the prototype pro-inflammatory signaling pathway (nuclear factor kappa B (NF-κB)). SCFAs also have roles in regulating inflammation and cancer [53].

Experimental studies have proven that regular consumption of fiber (especially the soluble fraction) in the optimal amount, i.e., about 30 g per day, is positively correlated with the production of butyrate by multiple bacteria, including *Faecalibacterium prausnitzii*, and members of the genus *Clostridium* and the genera *Butyrivibrio* and *Eubacterium* [54]. Adversely, in obese subjects with non-alcoholic fatty liver disease (NAFLD), a rapid rearrangement in the GM composition and, in consequence, in SCFAs production takes place within 24 h of reduction of carbohydrate (including DF) intake to 30 g/day [49].

A previous study has shown that a diet low in MACs contributes to a decrease in numerous specific bacterial taxa and reduces diversity, and this effect persists even after the reintroduction of MACs (Table 1) [40]. Since SCFAs regulate various pathways, the reduction in their production as a consequence of inadequate dietary fiber intake acts to deregulate energy balance and lipid and carbohydrate metabolism. Additionally, under conditions in which dietary polysaccharides are absent, GM use host mucus glycans as an alternative energy source. A study conducted by Desai et al. [55] using animal models showed that dietary fiber deprivation, and in consequence, erosion of the colonic mucus barrier, promoted greater epithelial access and enhanced inflammation and susceptibility to pathogens. A low intake of DF during a high-protein/low-CHO diet also shifts the GM metabolism toward the utilization of dietary and endogenously supplied proteins, causing elevated levels of cytotoxic and pro-inflammatory metabolites (such as branched-chain fatty acids, ammonia, amines, N-nitroso compounds, p-cresol, sulfides, indolic compounds, and hydrogen sulfide) [56].

#### 2.2.2. Proteins

Proteins are an essential component of the human diet, customarily providing 10–20% of daily energy requirements. Products of protein GM degradation constitute a broad group of compounds, which include SCFAs, indoles, amines, phenols, thiols, hydrogen sulfide, CO_2_, and H_2_ [51]. The effects of dietary protein on GM have been observed in several studies. The first study in this area described that individuals who consumed a high beef diet had lower counts of *Bifidobacterium adolescentis* and increased abundance of *Bacteroides* and *Clostridia* when compared to those in a meatless diet counter group [57]. This study initiated a series of interventional and observational studies that described the effects of diets with different protein contents (low protein, high protein diet) and various sources of protein in the diets on the establishment of the GM (Table 2) [58].

A majority of previous studies have concluded that protein consumption positively correlates with overall microbial diversity, but there are significant differences in bacterial enterotypes between individuals preferring animal-derived proteins versus plant-derived proteins. Consumption of animal-based protein is associated with an increase in counts of bile-tolerant anaerobes, such as *Bacteroides, Alistipes*, and *Bilophila* [59,60]. This relationship has also been described by Filippo et al. [61] in a study comparing GM of Italian children with that of children from Burkina Faso. The study compared equally-caloric diets with a different proportion of animal protein and carbohydrate (DF) and revealed that despite individuals on a high-protein and low CHO diet losing weight, their GM changed unfavorably—the count of *Roseburia* and *Eubacterium rectale* and proportion of butyrate decreased [62]. The previous study also revealed that a high-protein diet affected the increase in *Streptococcus*, *E. coli/Shigella*, and *Enterococcus* (5.36-fold, 54.9-fold, and 31.3-fold, respectively) and a decrease in beneficial *Faecalibacterium prausnitzii* (by 3.5-fold) and *Ruminococcus* (by 8.04-fold) [63].

#### 2.2.3. Lipids

Dietary lipids from plants and animals are a broad group of compounds that are a reserve source of energy for the human body and help determine its proper growth and development. The group encompasses fatty acids, including saturated (SFA), monounsaturated (MUFA), and polyunsaturated (PUFA) fatty acids, their derivatives, including mono-, di-, and triglycerides, and phospholipids, as well as sterols, such as cholesterol [69]. In a traditional, healthy diet, 30% of fat content in balanced proportion SFA:MUFA:PUFA is recommended. Both the quantity and the quality of dietary lipids play a role in GM establishment (Table 3). In observational studies, it has been described that a diet enriched in MUFA and PUFA increases the Bacteroidetes:Firmicutes ratio and elevates numbers of lactic acid bacteria—*Bifidobacteria* and *Akkermansia muciniphila.* Conversely, SFA (specifically long-chain saturated fatty acids) promotes the growth of *Bilophila* and *Faecalibacterium prausnitzii* and causes a decline in numbers of *Bifidobacterium*, *Bacteroidetes*, *Bacteroides*, *Prevotella*, *Lactobacillus* ssp. [70,71,72]. The diet high in SFA influences the secretion of bile acid and boosts levels of bile acids in the intestine, including increased production of hydrophobic secondary bile acids (e.g., deoxycholic acid—DCA). DCA leads to changes in the composition and structure of GM. Elevated DCA has been described as a factor promoting atherosclerosis, diabetes, and other cardiometabolic diseases [73]. Additionally, it has been considered that SFA as an essential component of the lipid portion of lipopolysaccharide (LPS) of pathogenic bacteria activates an inflammatory cascade via toll like receptor 4 (TLR-4) [73]. The interventional study has provided evidence that although both a high-SFA diet or a diet with a high n-6 to n-3 PUFA ratio promote weight gain, only increased SFA consumption is related to insulin resistance, visceral adiposity, and intestinal permeability [74]. The differences between mice fed an HFD and n-6 PUFA diet or LFD and n-3 PUFA diet are also documented. Caesar et al. [75] compared the gut microbiota of mice fed with an HFD based on lard (SFA) or fish oil (n-3 PUFA) and noticed that diversity and counts of commensal *Akkermansia muciniphila, Lactobacillus,* and *Bifidobacterium* were higher in the mice that consumed fish oil. Additionally, in the mice that consumed lard, it was observed that the Toll-like receptor 4 (TRL-4) was activated, and hence there was increased inflammation in white adipose tissue (WAT). A study addressing the effect of nut consumption on shaping the microbial community has demonstrated that a high amount of walnuts in the diet (43 g/d) result in significant changes in composition and diversity in the GM by enhancing *Ruminococcaceae* and *Bifidobacteria* and decreasing *Clostridium sp. cluster XIVa* species [76]. Additionally, significant changes have been observed in some bacterial enzyme activities (increased β-galactosidase activity and reduced fecal β-glucuronidase, nitroreductase, and azoreductase activities) in response to the consumption of nuts [53]. Similarly, the beneficial effects of PUFA-rich nuts on GM have been observed in other studies [77,78]. The different microbial enterotypes have been described in the previous study regarding diet with milk fat (increased pro-inflammatory *Bilophila wadsworthia*) [79]. It should be noted that a systematic review conducted by Wolters et al. [80] within the MyNewGut project concluded that results of available interventional studies in humans did not suggest strong effects of different amounts and types of dietary fat on the intestinal microbiota composition. They noticed that high consumption of fat and SFA might unfavorably affect microbiota richness and diversity, and high MUFA diets might decrease total bacterial abundance, whereas PUFA probably did not affect GM richness and diversity [80].

#### 2.2.4. Other Dietary Components and GM

A beneficial microbial community in the gut is influenced by several additional but essential factors, such as intake of probiotics, the content of bioactive substances, such as polyphenols, in the diet, dietary supplementation of vitamins and minerals, and abuse of alcohol/tea/coffee/salt in the diet.

Dietary polyphenols include a broad group of substances (such as catechins, flavonols, flavones, anthocyanins, phenolic acids) with well-described antioxidant properties. In everyday diets, people consume polyphenols mainly from fruits and vegetables, tea, cocoa products, and red wine. Previous studies have demonstrated that polyphenols have a beneficial effect on GM by increasing the abundance of *Bifidobacterium* and *Lactobacillus* and elevating the production of SCFAs. Additionally, a reduction in pathogenic bacteria—*Clostridium* species (*C. perfringens* and *Clostridium histolyticum*)—has been detected in response to polyphenols intake [81].

Probiotics are defined as “live microorganisms which, when administered in adequate amounts, confer a health benefit on the host” [82]. They occur naturally in fermented foods containing lactic acid bacteria (fermented milk, yogurt). The influence of probiotic-containing foods on the increase in total bacterial load, promotion of *Bifidobacteria* and/or *Lactobacilli* and/or *Streptococcus,* and reduction in *E. coli* and *Helicobacter pylori* have been widely described in previous reviews [83,84].

The evidence of selected studies concerning the influence of vitamins and minerals and the association between alcohol/salt/coffee/tea consumption and microbial community shaping is present in Table 4.

### 2.3. Dietary Pattern and Gut Microbiota

Although the influence of macronutrients and micronutrients on the composition and activity of the bacterial community is well described, it should be noted that the final diet-gut microbiota relationship is the result of their interaction. For this reason, it is reasonable to analyze the importance of specific dietary patterns and eating behaviors in establishing the intestinal microbiota.

#### 2.3.1. Vegetarian and Vegan Diet and GM

The GM composition (especially in the proportion of *Bacteroidetes*, *Prevotella,* and *Ruminococcus*) differs between individuals on vegan or vegetarian diets and omnivores, which has been well documented in previous studies [59]. Among the reasons for these changes are differences in bacteria directly consumed together with food, variations in pH, GI tract transit time, and differences in the number of substrates supplied for bacterial fermentation from the diet [59]. A plant-based diet naturally rich in DF, especially MACs, has a beneficial effect on the microbiome by inducing the development of more diverse and stable microbial systems. A comparative study describing the GM composition of the children from Burkina Faso consuming traditional plant-based, high fiber diet versus European children consuming a Western diet has shown profound differences [61]. GM in Burkina Faso children is characterized by significantly higher diversity with enrichment of *Bacteroidetes* and depletion of *Firmicutes* and the presence of the genera *Prevotella* and *Xylanibacter* (responsible for cellulose and xylan fermentation), which are absent in the gut of European children. It has been reported that *Prevotella*-dominated microbiota produces 2–3 times more propionate than the *Bacteroides*-dominated microbiota, and the amounts and ratios of the SCFA in both the enterotypes are different [103]. Additionally, in children from Europe, GM patterns include a higher abundance of potentially pathogenic *Shigella* and *Escherichia* than in the Burkina Faso group. The lower presence of butyrate-producing bacteria on a low fiber and high meat diet, through negative changes in colonic pH, promote the growth of pathogenic bacteria [54]. In 2017, De Filippo et al. analyzed microbial differences in small groups of children living in Burkina Faso (*n* = 11), of two groups of children living in different urban settings (Nanoro town, *n* = 8; Ouagadougou city, *n* = 5), and of a group of Italian children (*n* = 13) [104]. They observed that the introduction of animal-products (rich in fat and simple sugars) to the traditional African diet (rich in cereals, legumes, and vegetables) led to changes in microbiota profiles, but the microbiota of rural children retained a geographically unique bacterial reservoir (*Prevotella, Treponema*, and *Succinivibrio*), capable of degrading fiber and polysaccharides from vegetables. In children living in urban areas, these bacterial species were progressively outcompeted by bacteria more suited to the metabolism of animal protein, fat, and high sugar foods, similarly to Italian children. As a consequence, among the population of children from highly urbanized areas, a progressive reduction of SCFAs was observed. Significant differences in GM patterns have also been described between traditionally living Hadza (Tanzania) people and Italian industrialized society, with the former having the advantage of greater diversity and richness [105].

In addition to the above-described differences in the number of *Prevotella* bacteria, the vegetarian/vegan enterotypes differ from the omnivore enterotype by the number of *Bacteroides* and *Ruminoccosus* phyla. It has been reported that long-term animal-based diet positively correlates with the abundance of bile-tolerant microorganisms (*Bacteroides*, *Alistipes*, *Bilophilia*) and negatively with genera *Firmicutes*, which are able to degrade dietary plant polysaccharides (*Roseburia*, *Eubacterium rectale*, and *Ruminococcus*) [36]. An abundance of *Bacteroides* in response to a Western diet—high in animal products and low in fiber from fruits and vegetables—has been previously described [106].

Both animal and plant-based diets influence the *Ruminococcus* enterotype. *Ruminococcus* is responsible for the degradation of complex CHO (cellulose, RS) and plays a role in the conversion of animal-derived choline to trimethylamine (TMA). Previous studies have reported that an abundance of *Ruminococcus* is positively correlated with butyrate production, lower endotoxemia, lower arterial stiffness, and lower BMI [59].

In a study addressing the associations of short-term dietary changes, long-term nutritional habits, and lifestyle with gut microbiota, Klimenko et al. [107] reported that a higher intake of vegetables and fruits in the diet was associated with higher GM community diversity, but this effect was dependent on the initial microbiota state. Similarly, significant differences in the composition and diversity of the microbial community have been noticed in several studies [108,109,110,111].

In intervention studies involving the enrichment of the diet with fruit, vegetables, or other high-fiber products (such as products derived from whole grains), a favorable change in the composition and diversity of biota has been observed. A Danish study involving 123 non-obese and 169 obese subjects has shown that higher fruit and vegetable intake is associated with a high number of gut microbial genes that determine optimal bacterial richness. This study revealed that a calorie-restricted diet with increased fiber intake for six weeks led to an increase in microbial gene richness in the low microbial gene count (LGC) group, which remained significantly different from that of the high microbial gene count (HGC) group. The observed changes in microbial gene count affected the reduction of body fat and improvement in lipid and insulin levels and insulin sensitivity [60]. Evidence also suggests that GM established by a plant-based diet promotes lower production of secondary bile acids (DCA) and TMA and, therefore, reduces the risk of intestinal barrier dysfunction, inflammation, and liver cancer [112].

In conclusion, the available literature provides evidence that vegetarian and vegan diets are effective in promoting the optimal diversity and richness of beneficial bacteria (e.g., by increasing *Prevotella* taxa) and reduction in harmful metabolites (DCA, TMA), hence supporting both human GM and overall health.

#### 2.3.2. Mediterranean Diet (MD) and Gut Microbiota

Adherence to a Mediterranean diet (MD) has been linked to a reduced incidence of obesity and metabolic syndrome and decreased mortality and morbidity in patients with CVD [113,114]. Growing experimental and clinical evidence also suggests that an MD contributes to a beneficial GM pattern. The traditional MD is characterized by a high content of polyphenol-rich products (extra-virgin olive oil, red wine, vegetable, grains, legumes, whole-grain cereals, nuts), a beneficial proportion of fatty acids (high MUFA and PUFA, low SFA), and low consumption of processed meat and refined sugars. A raft of previous studies has revealed that this eating pattern could positively affect gut microbial communities. Mitsiu et al. [115] described that individuals with higher Mediterranean diet scores had lower *Escherichia coli* counts, a higher *Bifidobacteria* to *E. coli* ratio, increased levels and prevalence of *Candida albicans,* and a higher molar ratio of acetate compared with subjects with low adherence. De Filippis et al. [116] reported that high-level consumption of plant foodstuffs consistent with an MD was associated with a beneficial microbiome-related metabolomic profile (higher *Prevotella* and certain fiber-degrading Firmicutes profiles, higher SCFAs production) in subjects ostensibly consuming a Western diet [116]. The increased intestinal SCFAs level in subjects on the MD diet is determined by high consumption of vegetables, fruits, and legumes, which are rich sources of complex and insoluble fiber—the major substrates for microbial production of SCFAs. Another study conducted in patients from the CORDIOPREV study has shown that consumption of a Mediterranean diet or a low-fat diet might partially restore the gut microbiome dysbiosis in obese patients with metabolic syndrome [117]. Positive associations between higher adherence to an MD diet and increased levels of the butyrate-producing species—*Faecalibacterium prausnitzii* and *Clostridium* cluster XIVa—were observed by Gutiérrez-Díaz et al. [118]. Additionally, they detected that lower adherence to the MD was linked with higher urinary trimethylamine N-oxide (TMAO) levels. Similarly, in the PREDIMED study, it has been noted that concentrations of five metabolites in the choline pathway (TMAO, betaine, choline, phosphocholine, and α-glycerophosphocholine) in a group of individuals assigned to a Mediterranean diet are lower than those of a low-fat control diet group. The lower level of TMAO observed in individuals on an MD arises from the fact that in comparison to the WD, MD is characterized by significantly less (over 50% reduction) consumption of products (eggs, red meat, cheese) containing choline and l-carnitine, which are metabolized to TMA.

In conclusion, strong evidence has accrued that MD beneficially modulates gut microbiota by increasing the abundance of *Bacteroidetes, Clostridium* cluster XIVa, *Faecalibacterium prausnitzii, Lactobacilli,* and *Bifidobacteria* and decreasing the abundance of Firmicutes. This diet pattern positively affects the diversity and activity of various gut bacteria and increases the SCFAs level, and hence improves host metabolism. The mechanisms involved in establishing a beneficial GM in response to the MD diet are the effect of prebiotics on DF, the positive effect of polyphenols and n-3 PUFA, and a low intake of processed food.

#### 2.3.3. Western-Style Diet (WSD) and Gut Microbiota

The Western-style diet (WSD) with its high intake of energy-dense, high fat/meat (and processed meat), high sugar, salt, alcohol, and high content of processed food is associated with an elevated risk of cardiovascular diseases. Numerous studies have also demonstrated the negative effects of this diet model on gut microbiota richness and function. Being high in animal-derived protein and low in vegetables and fruits, a WSD leads to a significant decrease in numbers of total bacteria and commensal *Bifidobacterium* and *Eubacterium* species. It is suggested that a WSD may be associated with irreversibly reduced microbial diversity and depletion of specific bacterial species because of their low content of MACs [106]. Characteristics of the WSD—the low intake of DF and increased consumption of fat (mainly SFA)—lead to increased penetrability and a reduced growth rate of the inner mucus layer and hence increase the susceptibility to infections [119]. The negative effects of a WSD on the intestinal microbiome can be attributed to not only a poor intake of DF and a high intake of fat and animal-derived proteins but also to a high content of ultra-processed food and harmful food additives (e.g., emulsifiers, non-caloric artificial sweeteners (NAS)) [120]. An analysis by Partridge et al. [121] suggested that emulsifiers contributed to alterations in the gut microbiota, with changes to the intestinal mucus layer, increased bacterial translocation, and an impact on the associated inflammatory response. The effect of ultra-processed food on gut microbiota deregulation was also described by Martinez et al. [122], Viennois, and Chassaing [123], Suez et al. [124]. A study conducted by Chassaing et al. [123] on the influence of synthetic dietary emulsifiers polysorbate 80 (P80) and carboxymethylcellulose (CMC) on the gut microbiota revealed that both P80 and CMC affected deconstruction in the GM and led to increased expression of bacterial inflammatory molecules, such as flagellin [125]. Suez et al. [124] described that high consumption of commonly used non-caloric artificial sweeteners (NAS) promoted glucose intolerance through compositional and functional alterations to the GM. The role of NAS (such as sucralose, acesulfame-K, neotame) in deregulating the gut microbiome and promoting chronic inflammation has also been highlighted in several other studies [126,127,128].

In summary, there is strong evidence that the Western-style diet causes disorders of the microbiota, intensifies the chronic inflammatory process, and consequently leads to the development of metabolic disorders and cardiovascular diseases.

#### 2.3.4. Other Dietary Pattern and Dietary Habits and GM

The growing interest in describing the relationship between diet-dependent alteration of the GM and host metabolism has resulted in several studies, assessing the impact of different alternative nutritional models or everyday dietary habits on the establishment of the intestinal bacterial ecosystem.

Several studies have demonstrated that a gluten-free diet (GFD) causes a reduction in *Bifidobacterium* spp. and an increase in *Enterobacteriaceae* and *Escherichia coli* [129,130]. A preliminary study, conducted by Sanz et al. [131], showed that the abundance of healthy gut bacteria decreased, while counts of unhealthy bacteria increased in parallel to reductions in the intake of polysaccharides after following the GFD for more than one month. Even a short-term GFD followed for four weeks has affected the microbiota, with a reduction in family *Veillonellaceae* (class *Clostridia*) being noticed [132]. Similarly, significant changes in the composition of the microbiota and deregulation in its functions have been observed in people following other alternative dietary patterns, including the ketogenic diet [133,134], paleo diet [135,136], or intermittent fasting [137,138] (Table 5).

Currently, it is generally considered that irregular eating habits, such as skipping breakfast, having dinner late, and late-night eating, contribute to obesity and other metabolic disorders. Increasing evidence suggests that these improper diet habits may also result in deregulation in the composition and physiological function of GM by negatively influencing the circadian rhythm [19].

## 3. Gut Microbiota and Metabolic Disorders

### 3.1. Gut Microbiota Composition in Metabolic Disorders

One of the most significant public health problems today is obesity, defined as a chronic accumulation of lipids in fat tissue. Obesity is determined by many factors, but the most frequent cause is a disbalance between the consumption of energy from food and drinks and energy expenditure. Growing evidence suggests that the disbalance may be a consequence of dysregulated gut microbiota [142]. In the context of the development of obesity, the bacterial microflora participates in many relevant areas, including digestion and absorption of nutrients, energetic homeostasis (energy harvest), maintaining tightness of the intestinal epithelium, metabolism of carbohydrates, lipids, and bile acids, regulating intestinal motility, and regulating the immune and hormonal system.

The first studies concerning the impact of GM diversity and activity on metabolic disorders, particularly obesity, were based on findings from animal models. It has been observed that germ-free mice (especially raised mice devoid of all microorganisms) have a lower fat mass content than conventional mice, even though germ-free mice consume 29% more food [143]. Additionally, Bäckhed et al. [143] found a 60% increase in body fat mass and insulin resistance in adult germ-free mice within 14 days after colonization with GM from conventionally raised animals, despite having a reduced food intake [143]. It has also been noted that germ-free mouse with the obesity-resistance phenotype fed a high-fat diet (HFD) consumes fewer calories and has increased excretion of lipids, while an HFD in mice with normal microbiota contributes to weight gain, obesity progression, worsening in insulin sensitivity and inflammatory changes in the small intestine [144].

Upon investigating the cause of these phenomena, it has been discovered that GM composition differs between slim and obese mice. In comparison with lean mice, the microbiota of mice with obesity is characterized by a lower abundance of Bacterioidetes and a higher number of Firmicutes [142]. Similarly, differences in GM composition have been observed between obese and lean human subjects, with an elevated Firmicutes to Bacteroidetes ratio, and a higher proportion of Actinobacteria in obese subjects, and a reduced diversity and altered representation of bacterial genes and metabolic pathways [145]. Riva et al. [145] reported that obesity was associated with elevated levels of Firmicutes, such as *Ruminococcaceae*, and depleted levels of Bacteroidetes, such as Bacteroidaceae and Bacteroides. The Danish study observed that individuals with lower microbial gene count (LCG) were characterized by obesity, insulin resistance, adiposity, dyslipidemia, and a worse inflammatory status [146]. The obese gut microbiota phenotype is characterized by genes that participate in energy harvest and metabolism, specifically genes responsible for expressing enzymes that breakdown potentially indigestible complex plant polysaccharides to produce SCFAs. The systematic review has included seven human clinical studies with a total of 246 obese cases and 198 healthy controls and has shown that subjects with obesity have higher fecal levels of acetate, propionate, and butyrate SCFAs, as compared with lean controls [147]. The higher microbial capability to oxidize CHO in obese and the influence of SCFA on body fat partitioning were noticed by Goffredo et al. in children and adolescents [148]. They reported that a higher concentration of fasting plasma SCFA was associated with the percent total body fat, visceral fat, and positively predicted the changes in adiposity. What’s more, the level of SCFAs was correlated with rates of de novo lipogenesis [148].

Dysregulated GM composition and functionality have also been described in patients with type 2 diabetes. Karlsson et al. [149] noticed that individuals with T2D, which could be considered a complication of obesity, had a lower abundance of fiber-degrading bacteria. Larsen et al. [150] described the reduction in phyla Firmicutes and *Clostridia* and the increase in Bacteroidetes to Firmicutes ratio, as well as an increased *Bacteroides-Prevotella* ratio in patients with T2D. In other studies, the decrease in butyrate-producing bacteria (*Akkermansia muciniphila*, *Roseburia intestinalis,* and *Faecalibacterium prausnitzii*), *Haemophilus* and *Lactobacillus* and an increase in *Desulfovibrionaceae* spp and various opportunistic pathogens (*Clostridium* spp., *Bacteroides caccae*) have been observed [151,152,153]. The significantly elevated abundance of Proteobacteria had been described by Amar et al. [154] as an independent marker of the risk of insulin resistance and diabetes in humans. It had also been observed that dysregulation in gut microbiota composition leading to impairment in gut integrity and metabolic endotoxemia was one of the leading factors in insulin resistance and the development of T2D. In an animal model, intravenous administration of lipopolysaccharides is a triggering factor, leading to weight gain and insulin resistance [155]. The same mechanism involving GM dysbiosis, an elevated level of circulating LPS, and the promotion of T2D has been elucidated in humans [156]. On the other hand, intervention with the use of oligofructose and long-chain inulin or with B-glucans has shown that along with an improvement in GM, positive changes in glucose metabolism are also noticed [157]. Nonetheless, microbiota functionality in subjects with T2D differs from that of people with normal weight. There has also been observed enriched membrane transport of sugars and branched-chain amino acids, increased sulfate metabolism products, reduced butyrate synthesis, and changes in the metabolism of cofactors and vitamins [151].

Another metabolic disorder associated with obesity is dyslipidemia. Dyslipidemia is defined as an elevation of circulating triglycerides (greater than 1.7 mmol/L (150 mg/dL)) or a reduction in circulating high-density lipoprotein (HDL) (less than 1.0 mmol/L (40 mg/dL) in men and less than 1.3 mmol/L (50 mg/dL) in women), which contribute to the development of atherosclerosis and cardiovascular disease [158]. Previous studies have observed that GM in individuals with impaired lipid metabolism differs from that of healthy humans. Cotillard et al. [60] noted that the reduced microbial richness commonly observed in obese patients was also linked with increased total serum cholesterol and serum triglycerides (TG). Similarly, Le Chantelier noticed higher levels of TG in individuals with low microbial gene counts vs. those with high microbial gene counts [146]. Previous studies have not clearly defined the microbiota pattern characteristic of patients with dyslipidemia. By analyzing the composition of the microbiota of patients with symptomatic atherosclerosis, Karlsson et al. [149] observed a higher abundance of the genus *Collinsella* and a lower abundance of *Eubacterium* and *Roseburia* compared with healthy controls. Deregulation in gut microbiota composition (decreased phylum Bacteroidetes and increase in *Lactobacillales*) in patients with coronary artery disease (CAD) has also been revealed [159]. Additionally, a study with mice fed a high-fat high-cholesterol diet and supplemented with *Lactobacillus curvatus* and/or *Lactobacillus plantarum* has shown that commensal probiotic bacteria play important roles in normalizing lipid metabolism. In this study, *Lactobacillus curvatus* and/or *Lactobacillus plantarum* reduced cholesterol in plasma and liver and reduced the accumulation of hepatic triglycerides [160]. A comparable, positive anti-obesity and lipid-lowering effect of *Bifidobacterium* spp has been described by An et al. in obese rats fed a high-fat diet [161]. As in the case of insulin resistance, the decrease in LPS concentrations in response to infusion of polymyxin B has improved the markers of lipid metabolism and fatty hepatitis in mice [162].

### 3.2. The Mechanism Underlying Gut Microbiota-Related Metabolic Disorders

The alteration of microbiota composition and functionality may stimulate the development of metabolic disorders via several mechanisms involving increased energy harvest from the food, increased gut permeability leading to metabolic endotoxemia and its consequences, alterations in bile acid metabolism and G-protein-coupled bile acid receptor (FTR/TGR5) signaling, and influence of microbial metabolites: SCFAs, TMAO, indoles, LPS on various metabolic pathways.

#### 3.2.1. Role of SCFAs in Energy Harvest

Metabolic functions of SCFAs produced by microbial fermentation are essential in human energy homeostasis and, therefore, may significantly influence the development of obesity, insulin resistance, type 2 diabetes, and lipid disorders. SCFAs act as both an energy substrate and as signaling molecules, influencing lipogenesis, oxidation of fatty acids, fat storage, and gluconeogenesis [163]. SCFAs increase lipogenesis, i.e., they increase triglycerides, and inhibit the inhibitor of lipoprotein lipase in the small intestine, which results in inhibition of fatty acid release from triglycerides, and hence promotes the cellular uptake of triglycerides, resulting in increased fat storage.

The SCFAs acetate, propionate, and butyrate, which are found at an approximate molar ratio of 60:20:20, respectively, are an essential energy source and source of nutrition for the intestinal epithelium. Because 95% of SCFA are absorbed in the intestinal lumen, their elevated circulating plasma concentration, which characterizes individuals with obesity, provides an additional energy source, and in consequence, promotes de novo hepatic lipogenesis. It has been estimated that the increased rate of SCFA production observed in obese humans contributes at least 10% of overall energy intake (up to approximately 200 kcal/day). This “energy storage hypothesis” has been confirmed in several studies in the human population [148,164,165].

SCFAs also act as signaling molecules and specific ligands for G protein-coupled receptors. GPR41 and GPR43 regulate the secretion of anorectic hormones peptide YY (PYY) and production of insulinotropic intestinal hormones (incretins; glucagon-like peptide 1—GLP-1, GLP-2). These are currently regarded as the main factors determining postprandial insulin secretion and pancreatic β-cell function [106]. GLP-1 is known to have beneficial effects on glucose metabolism, while GLP-2 has been shown to improve the integrity of the intestinal epithelial tight junction [155]. PYY and its co-secreted GLP-1 play an important role in the endocrine regulation of appetite and satiety. Increased PYY secretion normally inhibits gut motility, accelerates the intestinal passage, and, therefore, reduces calorie extraction from the diet. Gpr41 also mediates SCFA-induced synthesis of leptin, an adipocytokine with pleiotropic effects on appetite and energy metabolism [166]. Stimulation of Gpr43 by SCFAs increases the differentiation of peroxisome proliferator-activated receptor-gamma (PPARγ) and also reduces lipolysis, which leads to the development of adipose tissue accumulation. A recent study has shown that *Gpr43−/−* mice are resistant to diet-induced obesity and insulin resistance, at least partly due to Gpr43-regulated energy expenditure [167]. Furthermore, gut microbiota also affects the energy harvest by activating carbohydrate responsive element-binding protein (ChREBP) and sterol responsive element-binding protein (SREBP-1). In consequence, increased de novo lipogenesis is promoted in adipose tissue and liver [168]. The SCFAs are involved in lipid oxidation and energy expenditure also by their influence on the activation of the AMP-activated protein kinase (AMPK) in adipose tissue and skeletal muscles and regulation of peroxisome proliferator gene (PPAR). AMPK activates acetyl-CoA carboxylase (ACC) and carnitine-palmitoyltransferase I (CTP1)—two enzymes responsible for mitochondrial fatty acid oxidation. SCFA-induced activation of PPARγ modulates lipid metabolism through increased energy expenditure and decreased liver triglyceride accumulation. It has been described that germ-free mice have higher levels of phosphorylated AMPK in muscle and liver than conventional mice, and this determines their better control of energy expenditure [143].

However, there have been some mixed results concerning the relationship between SCFAs and obesity. For example, in some interventional studies, a positive correlation between SCFA supplementation and weight loss and insulin sensitivity by stimulating PYY and GLP-1 have been observed [169]. In light of these studies, the question arises whether the role of SCFAs in the development of obesity is dose-dependent (SCFAs at pharmacological doses vs. SCFAs in amounts produced during fermentation) and whether it is likely that an elevation in SCFAs in obese subjects might be a consequence, and not the cause, of an energy hyperalimentation, and might be a mechanism of defense against energy accumulation. It is suggested that despite a high level of SCFAs in obese individuals, they have become unresponsive to the regulatory effect of SCFAs on appetite and energy expenditure.

#### 3.2.2. Metabolic Endotoxemia

The additional mechanism linking dysregulation of the GM and metabolic disorders is increased intestinal permeability, which induces metabolic endotoxemia and initiates an inflammatory cascade. In physiological conditions, the integrity of enterocytes is maintained by tight junctions, gap junctions, and desmosomes. In the dysbiosis state, there is an abnormal synthesis of proteins stabilizing tight junctions, such as zonulin-1 and occludin. As a consequence, the translocation of allergens and other toxic substances, including bacterial lipopolysaccharides LPS (endotoxin derived from the cell wall of gram-negative bacteria), from the intestinal lumen to the bloodstream is promoted. Cani et al. [170] in an animal model study noted that in response to an HFD, the plasma LPS level increased. The higher LPS concentration is associated with the development of obesity and insulin resistance [171]. On the other hand, an increased LPS level and obesity have been observed in mice fed a control diet after LPS infusion. In humans, it has been proven that patients with obesity, insulinresistance (IR), or type 2 diabetes (T2DM) have increased plasma concentrations of LPS compared to a healthy counter group [155].

The high concentration of LPS in circulation activates Toll-like receptors 4 (TLR-4) and nucleotide-binding oligomerization domain (NOD) expression and initiates low-grade inflammation, which ultimately contributes to obesity and development of metabolic disorders. LPS also activates inflammasomes, leading to maturation of interleukin-1β (IL-1β) and IL-18, which are proinflammatory cytokines. Previous studies have reported an association of the NOD-, LRR- and pyrin domain-containing protein 3 (NLRP3) inflammasome with metabolic disorders, such as obesity and/or type 2 diabetes and insulin resistance [172].

The dysbiosis is closely associated with disturbances in T reg and T helper proportion. The negative influence of endotoxemia on obesity and pathogenesis of metabolic disorders is also implicated in inflammatory cytokine overproduction in adipocytes, reduced adiponectin synthesis, increased secretion of leptin and resistin, and in consequence, elevated insulin-like growth factor—IGF-1 and IGF-2. Increased circulation of LPS has linked also with exacerbating triglycerides’ accumulation in adipose tissue and muscle because it suppresses the fasting-induced adipose factor (FIAF). FIAF or angiopoietin-like protein 4 is a circulating lipoprotein lipase (Lpl) inhibitor produced by the intestine, liver, and adipose tissue. A previous study has demonstrated that introducing a gut microbiota to germ-free mice suppresses expression of FIAF in the gut epithelial cells and, therefore, leads to a higher adipocyte Lpl activity. As a consequence, there is an increased deposition of triglycerides in adipocytes and greater fat storage. A study in germ-free knock-out FIAF mice has shown that high-fat/high-carbohydrate diet affects obesity, which proves that FIAF plays a pivotal role in microbial regulation of energy harvest [143].

#### 3.2.3. Bile Acid Metabolism

Primary bile acids, cholic acid, and chenodeoxycholic acid are synthesized in the liver as final degradation products of endogenous cholesterol. Primary bile acids play a critical role not only in the digestion and absorption of dietary lipids and fat-soluble vitamins, but they also affect intestinal barrier permeability and the inflammatory response. As signaling molecules, they regulate energy metabolism and lipid and glucose metabolic pathways. The bile acids are stored in the gallbladder, but in response to food ingestion, they are excreted into the duodenum, where they participate in the emulsification of dietary lipids. About 95% of the total primary bile acids are reabsorbed from the ileum to the liver, and the remaining 5% is deconjugated, dehydroxylated, and epimerized to secondary bile acids [173] Several molecular mechanisms regulate the enterohepatic circulation of bile acids, but the most important receptors for bile acid synthesis are farnesoid X receptor (FXR) and Takeda G-protein-coupled receptor (TGR5), which regulate diverse metabolic pathways in the host [174]. Binding bile acids to FXR activates transcription of genes participating in the regulation of bile acid reabsorption, including IBABP (ileal bile acid-binding protein), organic solute and steroid transporter α and β (OSTα/β), a growth factor for fibroblasts (fibroblast growth factor 19—FGF19), and short heterodimer partner (SHP). By the induction of the FXR-dependent expression of SHP and FGF15/FGFR4, the bile acids maintain a balance between the synthesis of bile acids and entero-hepatic circulation of bile acids. Moreover, FXR is involved in regulating lipid metabolism, specifically very low density lipoprotein (VLDL) and triglyceride synthesis and utilization, and control of hepatic de novo lipogenesis [175]. Previous studies have described the relationship between bile acids and glucose metabolism, suggesting the role of FXR activation in reducing serum glucose and improving insulin sensitivity [176,177]. FXR via small heterodimer partner (SHP) and stimulation by FGF-19 increases glycogen synthesis and reduces gluconeogenesis [177]. A role of FXR activation in glucose-induced insulin transcription and secretion has been proposed [178]. Additionally, it has been shown that FXR also plays an anti-inflammatory role and maintains the integrity of the intestinal barrier, preventing bacterial translocation in the digestive tract.

Takeda G-protein-coupled receptor (TGR5) is expressed in brown adipose tissue and muscle. Similar to FXR, TGR5 not only participates in the regulation of bile acid metabolism but also is a factor triggering human energy homeostasis and glucose and lipid metabolism. TGR5 activation promotes energy expenditure by its influence on the conversion of inactive thyroxine (T4) to active tri-iodothyronine (T3). In addition, TGR5 signaling induces GLP-1 release from L-cells, resulting in improved liver function and enhanced glucose tolerance [175].

The relationship between the bile acids and the intestinal microbiota is bidirectional. Both bile acids are an essential modulator of the microbial intestinal community, yet the composition of GM affects the proportion and amount of synthesized bile acids. It has been proposed that the conversion of primary to secondary bile acids is dependent on GM, and the type of secondary bile acids produced is determined by diet and the condition of the GM [175]. On the other hand, because different bile acids affect FXR in different ways (as an agonist or antagonist), the dysregulation in GM may lead to the impairment of host metabolism. The role of FXR in changing the gut microbiota composition and hence increasing adiposity has been proven in several animal studies [176,177]. Parséus et al. [178] in a study with germ-free (GF) and conventionally-raised (CONV-R) wild-type and Fxr−/− mice fed an HFD showed that microbiota-induced weight gain, hepatic steatosis, and inflammation were dependent on FXR signaling. Jiang et al. [179] reported that treatment of obese mice with glycine-β-muricholic acid (Gly-MCA) inhibited FXR signaling in the intestine and, therefore, improved metabolic parameters.

#### 3.2.4. Trimethylamine N-Oxide (TMAO)

In the context of metabolic disorders and CVD, a strong relationship between gut microbiota dysbiosis and increased level of trimethylamine N-oxide (TMAO) has also been described. The gut microbiota metabolizes choline, phosphatidylcholine, and l-carnitine to trimethylamine (TMA). TMA is further transformed into trimethylamine N-oxide (TMAO), which may promote increased atherosclerosis through mechanisms related to lipid metabolism and inflammation. It has been reported that subjects with atherosclerosis have elevated levels of circulating TMAO compared with healthy controls [180]. It has also been noted that elevated levels of TMAO are strongly associated with type 2 diabetes. The link between the gut microbiota-initiated trimethylamine-N-oxide (TMAO)-generating pathway and obesity and the beiging of white adipose tissue were described by Schugar et al. [181]. Hepatic flavin monooxygenase 3 (FMO3) is involved in the oxidation of TMA to TMAO in the liver. The experimental study has demonstrated that knockdown or genetic deletion of FMO3 has an anti-obesity effect. The expression of FMO3 is regulated by bile acid-activated FXR, and this relationship shows how interconnected are the pathways, linking intestinal dysbiosis and the promotion of metabolic diseases [182].

#### 3.2.5. Tryptophan-Derived Metabolites

Tryptophan (Trp) is one of the essential aromatic amino acids supply from nutritional sources. About 4% to 6% of tryptophan is metabolized by GM into indican, indole or indole acid derivatives, skatole, and tryptamine [183]. The gut bacteria-derived indoles produced from tryptophan are a key modulator of host physiological and pathological pathways and hence may contribute to the cardiovascular, metabolic, and brain disorders [184]. The metabolism and biological effects of gut bacteria-derived indoles have been widely reviewed in several publications [184,185,186]. Indole propionic acid (IPA) is a metabolite produced by a small group of bacteria (especially *Clostridium sporogenes*) from dietary tryptophan. Previous studies have provided evidence that IPA plays an important role in intestinal barrier stabilization, antioxidant defense, and has neuroprotective effects [184]. The study conducted by Dodd et al. [187] in genobiotic mice inoculated with wild-type *C. sporogenes* showed that serum level of IPA was related to intestinal permeability and immune activation in a pregnane X receptor (PXR)-dependent fashion. The previous study comparing differences in gut microbiota-dependent metabolites between GF and conventionally raised mice contrary to mice fed an high-fat diet (HFD) or low-fat diet (LFH) revealed that indole-3-acetate (I3A) and tryptamine were depleted under an HFD. Both metabolites, especially I3A, modulate inflammatory responses by decreased fatty-acid- and LPS-stimulated release of pro-inflammatory cytokines and chemokine [188].

#### 3.2.6. The State-of-Art: Microbiome-Based Treatment

The growing body of reports describing the diet-GM-host health relation has resulted in researches concerning the use of microbial remodeling strategies in the treatment of metabolic diseases. The recent systematic review on six RCTs study conducted by Tenorio-Jiménez et al. [189] evaluated that the probiotics intake in patients with MetS might improve some clinical parameters, such as body mass index, blood pressure, glucose metabolism, lipid profile, and inflammatory biomarkers, but these beneficial effects seem to be clinically non-relevant [189]. However, it should be emphasized that the available data are still insufficient to recommend the supply of probiotics and/or prebiotics to all patients suffering from metabolic disorders explicitly. The rationalization of the use of probiotic-based strategies in the treatment of obesity and co-existing metabolic disorders requires the use of targeted types and strains of bacteria or/and their combinations, as well as standardization of dose, time, and form of their supply. Identifying these elements seems to be critical to future research into the use of probiotics in the treatment of metabolic disorders. The personalized diet recommendation based on GM or using metabolite supplementation also may successfully modify various metabolic pathways and hence affect improvement in human health [190].

## 4. Conclusions

Long term dietary pattern appears to have a pivotal influence on GM composition and functionality and hence on host metabolism. Proper nutrition, defined as a calorie-balanced diet, with adequate fruit and vegetable intake, rich in dietary fiber, with healthy fats (MUFAs and PUFAs), and a predominance of plant-derived proteins seem to be the best way to promote GM diversity and activity. Previous studies in animals and humans have observed that human GM differ between obese and lean individuals or in individuals with metabolic disorders. The deregulation in gut microflora predisposes to impairment of host metabolism in many ways, including the bacterial influence on energy harvest from food, damage in intestinal epithelium tightness leading to metabolic endotoxemia, and alteration in the metabolism of carbohydrates, lipids, and bile acids. Therapies targeting GM dysbiosis may, in the future, be a promising tool in support of the treatment of patients with several diseases, such as obesity, dyslipidemia, insulin resistance, or type 2 diabetes. In-depth knowledge of connections between GM establishment, metabolism, and human health will allow targeting microbiome reprogramming for the prevention and treatment of metabolic diseases in the future. The manipulation of microbiota-derived metabolites and their pathways can be helpful in exploring novel, individualized, and efficient treatment modalities for various human disorders.

## Figures and Tables

**Table 1 nutrients-12-01096-t001:** Influence of digestible and non-digestible carbohydrates on gut microbiota (GM).

Reference	Study Type	Population	Dietary Sources of Carbohydrate	Influence on Gut Microbiota
Fehlbaum et al. 2018 [40]	in vitro study	screening platform (i-screen) inoculated with adult fecal microbiota	a different source of dietary fiber (DF): FOS (chicory root), inulin (chicory root), alpha-GOS (peas), beta-GOS (lactose), XOS-C (corn cobs), XOS-S (sugar cane fiber), and β-glucan (oat flour)	β-glucan induced ↑ *Prevotella* and *Roseburia* and ↑ SCFA propionate production. Inulin and GOS, XOS induced ↑ *Bifidobacteria* all DF had a prebiotic activity with β-glucan being dominant
Do et al. 2018 [41]	animal experimental study	eight-weeks-old male C57BL/6J mice (*n* = 36)	normal diet (ND), HGD (high glucose diet), HFrD (high fructose diet), or HFD (high-fat diet) for 12 weeks	HGD and HFrD caused ↑ *Akkermansia*, ↓ microbial diversity (↓ *Bacteroidetes*, ↑ *Proteobacteria*) vs. HFD group
Sen et al. 2011 [42]	animal experimental study	Sprague-Dawley rat (*n* = 12)	HF/HSD, LF/HSD, or control low-fat/low-sugar diet (LF/LSD) for 4 weeks	HF/HSD and LF/HSD-fed caused gut microbiota dysbiosis (↑ *Clostridia* and *Bacilli*, ↓ *Lactobacillus* spp), ↓ bacterial diversity, ↑ *Firmicutes/Bacteriodetes* ratio LF/HS ↑ *Proteobacteria* (*Sutterella* and *Bilophila)* HF/HSD and LF/HSD increased LPS
Whelan et al. 2005 [43]	a randomized, double-blind, crossover trial	healthy subjects (*n* = 10)	standard enteral formula vs. formula supplemented with FOS (5.1 g/L) and fiber (8.9 g/L) as a sole source of nutrition for 14 days	FOS/fiber formula led to ↑ *Bifidobacteria* and ↓ *Clostridia* and induced higher concentrations of total SCFA, acetate, and propionate
Martinez et al. 2010 [44]	a double-blind, crossover trial	heathy human (*n* = 10)	crackers containing either RS2 (resistant starch type 2), RS4, or native stRS types 2 (RS2) and 4 (RS4) for 3 weeks	RS4 but not RS2 induced significantly, reversible ↑ *Actinobacteria* and *Bacteroidetes* and ↓ *Firmicutes*. RS4 induced ↑ *Bifidobacterium adolescentis* and *Parabacteroides distasonis*, RS2 induced ↑ proportions of *Ruminococcus bromii* and *Eubacterium rectale*
Davis et al. 2011 [45]	single-blinded study	healthy human subjects (*n* = 18)	GOS-containing products with four doses (0, 2.5, 5, and 10 g GOS) for 12 weeks	↑ *Bifidobacterium* (at the expense of *Bacteroides*) with a dose-dependent manner
Walker et al., 2011 [46]	randomized crossover trial	overweight adult men (*n* = 14)	HRSD (high in resistant starch diet), NPS (diet high in non-starch polysaccharides), WL (reduced CHO diet) vs. control diet for 10 weeks	HRSD ↑ *Ruminococcus bromii* and *Eubacterium rectale*
Francavilla et al. 2012 [47]	observational prospective study	infants with cow’s milk allergy vs. control (*n* = 28)	formula with no lactose for 2 months followed by an identical lactose-containing formula for an additional 2 months	↑ *Lactobacillus/Bifidobacteria* and ↓ *Bacteroides/Clostridia* ↑ SCFAs
Hald et al. 2016 [48]	randomized crossover study	adults with metabolic syndrome (*n* = 19)	a diet enriched with AX (arabinoxylan) and RS2 (resistant starch type 2) vs. low-fiber Western-style diet for 4-weeks	AX, RS2 caused ↓ total species diversity, ↑ heterogeneity of bacterial communities both between and within subjects, induced ↑ *Bifidobacterium*, ↑ total SCFAs, ↑ acetate, ↑ butyrate, ↓ isobutyrate, and ↓ isovalerate
Nicolucci et al. 2017 [38]	double-blind, randomized placebo-controlled trial	children (*n* = 30; 7–12 years) with overweight/obesity (>85th percentile of BMI) but otherwise healthy	oligofructose-enriched inulin (OI); 8 g/day; *n* = 22) diet vs. maltodextrin placebo diet (isocaloric dose, *n* = 20) for 16 weeks	↓ body weight z-score (3.1%), percent body fat (2.4%), percent trunk fat (3.8%), and IL-6, TG level in OI group ↑ *Bifidobacterium* spp. and ↓ *Bacteroides vulgatus* in the OI group
Mardinoglu et al. 2018 [49]	short-term intervention study	obese subjects with non-alcoholic fatty liver disease (*n* = 10)	isocaloric low-CHOs diet (30 g/d) with increased protein content by 14 days	rapid reduction (after 24 h) of fiber-degrading bacteria, ↑ *Lactococcus, Eggerrthella*, and *Streptococcus*↓ SCFAs level
Jones et al. 2019 [50]	observational prospective study	obese Hispanic adolescent (12–19 years) (*n* = 52)	the mean daily intakes of energy, fiber, protein, fat, CHO, sugars, and fructose assessment with the use of 24-h diet recall	high fructose in the diet was associated with ↓ *Eubacterium eligens* and *Streptococcus thermophilus*

Alpha-linked galactooligosaccharides (alpha-GOS), arabinoxylan (AX, beta-linked galactooligosaccharides (beta-GOS), carbohydrates (CHOs), dietary fiber (DF), HFD (high-fat diet), HFrD (high fructose diet), HGD (high glucose diet), HRSD (high in resistant starch diet), oligofructose-enriched inulin (OI), short chain fatty acid (SCFA), xylooligosaccharides from corn cobs (XOS-C), xylooligosaccharides from sugar cane fiber (XOS-S), normal diet (ND), a diet high in non-starch (NPS).

**Table 2 nutrients-12-01096-t002:** Influence of different sources of proteins on GM.

Reference	Dietary Sources of Proteins	Study Type	Population	Influence on Gut Microbiota
Meddah et al. 2001 [64]	whey protein, duration of study	in vitro study	simulator of the human intestinal microbial ecosystem (SHIME)	increased *Bifidobacterium* and *Lactobacillus* and decreased *Bacteroides fragilis* and *Clostridium perfringens* increase in acetic acid, CH_4_, and CO_2_ production, suggesting overgrowth of some anaerobic bacteria
Świątecka et al. 2011 [65]	glycated pea protein duration of study	in vitro study	batch-type simulator models imitating human intestinal conditions	increased *Bifidobacterium* and *Lactobacillus* increased levels of the SCFAs
Zhu et al. 2015 [66]	red meat (beef and pork), white meat (chicken and fish), and other sources (casein and soy) duration of study	animal experimental study	male Sprague-Dawley rats 3 wk old (*n* = 119)	protein type in diets had a significant effect on gut bacteria in the caecum white meat: higher *Lactobacillus* vs. red meat or non-protein diet chicken and fish proteins: higher *Firmicutes*, lower *Bacteroidetes* vs. other proteins soy protein: higher *Bacteroidetes* chicken protein: a greater abundance of *Actinobacteria* beef protein: higher *Proteobacteria*
Butteiger et al. 2016 [67]	soy protein vs. milk protein (MPI) duration of study	animal experimental study	6- to 8-week-old, male Golden Syrian hamsters (*n* = 32)	reduced abundance of *Bacteroides* and increased abundance of *Proteobacteria* *in MPI group*
Zhou et al. 2018 [68]	buckwheat protein duration of study	animal experimental study	male C57BL/6 mice (*n* = 27)	increased *Lactobacillus*, *Bifidobacterium,* and *Enterococcus* reduced *Escherichia coli*

**Table 3 nutrients-12-01096-t003:** Influence of lipids on GM.

Reference	Study Type	Population	Dietary Sources of Lipids	Influence on Gut Microbiota
Caesar et al. 2015 [75]	animal experimental study	Trif(−/−) and Myd88(−/−) mice (*n* = 30)	isocaloric diets (45% kcal fat) of the identical composition except for the source of fat—lard vs. fish oil for 11 weeks	increased *Bacteroides*, *Turicibacter*, and *Bilophila* in lard-fed mice, increased *Actinobacteria*, lactic acid bacteria, *Verrucomicrobia* in fish-oil-fed mice
Devkota et al. 2012 [79]	animal experimental study	Il10 −/− mice (*n* = 15)	milk fat (MF), lard fat (LF), or polyunsaturated fatty acids (PUFAs) test diets for 1 week	MF but not PUFAs promoted *Bilophila wadsworthia*
Bamberger et al. 2018 [76]	randomized, controlled, prospective, cross-over study	healthy Caucasian subjects (*n* = 194)	walnut-enriched diet (43 g/day) vs. a nut-free diet for 8 weeks	walnut consumption increased abundance of *Ruminococcaceae* and *Bifidobacteria* and decreased *Clostridium sp. cluster XIVa species*
Tindall et al. 2019 [78]	randomized, crossover, controlled-feeding trial	adults at CVD risk (*n* = 42)	2-week standard Western diet (SWD) run-in and three 6-wk isocaloric diets: containing whole walnuts (WD; 2.7% α-Linolenic acid (ALA)), a fatty acid-matched diet devoid of walnuts (WFMD; 2.6% ALA), replacing ALA with oleic acid without walnuts (ORAD; 0.4% ALA)	WD: the most abundant was Eubacterium eligens group, Lachnospiraceae, Lachnospiraceae, and Leuconostocaceae WFMD: the most abundant was Roseburia and Eubacterium eligens

α-Linolenic acid (ALA), cardiovascular disease (CVD), lard fat (LF), milk fat (MF), oleic acid diet (ORAD), polyunsaturated fatty acids (PUFAs), standard Western diet (SWD), whole walnuts diet (WD), walnuts free diet (WFMD; 2.6% ALA).

**Table 4 nutrients-12-01096-t004:** Influence of selected dietary factors on GM.

Factor	Reference	Study Type	Population	Diet	Influence on Gut Microbiota
Iron	Jaeggi et al. 2015 [85]	double-blind, randomized controlled trial	6-month-old Kenyan infants (*n* = 115)	home-fortified maize porridge (12.5 mg Fe/daily) for 4 months	adversely affected the GM, increased pathogen abundance and intestinal inflammation increased *Enterobacteria* (*Escherichia/Shigella*), *Enterobacteria/Bifidobacteria* ratio, *Clostridium*, *E. coli* increased fecal calprotectin
Iron	Zimmermann et al. 2010 [86]	randomized, double-blind, controlled trial,	6–14-year-old Ivorian children (*n* = 139)	iron-fortified biscuits, which contained 20 mg Fe/d, 4 times/wk as electrolytic iron or non-fortified biscuits for 6 months	iron: increased *Enterobacteria* and decreased *Lactobacilli*, increased mean fecal calprotectin concentration
Iron	Mahalhal et al. 2018 [87]	animal experimental study	adult female C57BL/6 mice with/without DSS-induced colitis (*n* = 130)	chow diets containing either 100, 200, or 400 ppm iron	dietary iron at 400 ppm resulted in a significant reduction in the fecal abundance of *Firmicutes* and *Bacteroidetes* and increase of *Proteobacteria* and *Actinobacteria*
Vitamin D	Waterhouse et al. 2019 [88]	systematic review	mice study (*n* = 10) and human study (*n* = 14)	diets containing different levels of vitamin D (usually high versus low)	increase in *Bacteroidetes* in the low vitamin D diet
Vitamin D	Naderpoor et al. 2019 [89]	a double-blind, randomized, placebo-controlled trial	26 vitamin D-deficient overweight or obese healthy adults (*n* = 32)	100,000 UI of cholecalciferol/d followed by 4000 IU/d or placebo for 16 weeks	higher abundance of genus Lachnospira, and lower genus Blautia in the supplemented group individuals with 25(OH)D >75 nmol/L had a higher abundance of Coprococcus and lower abundance of Ruminococcus compared to those with 25(OH)D <50 nmol/L
Vitamin D	Charoenngam et al. 2020 [90]	a randomized, double-blind, dose-response study	adults with vitamin D deficiency (*n* = 20)	600, 4000, or 10,000 IUs/day of oral vitamin D_3_ for 8 weeks	baseline serum 25(OH)D was associated with an increased relative abundance of *Akkermansia* and decreased abundance of *Porphyromonas* a dose-dependent increase in *Bacteroides* with a significant difference between the 600 IUs and the 10,000 IUs groups, and *Parabacteroides* with a significant difference between the 600 IUs and the 4000 IUs groups
Alcohol	Wang et al. 2018 [91]	animal experimental study	female BALB/c mice (6 weeks old) (*n* = 30)	bottle with increasing alcohol concentration (3%, 6%, 10%, *v*/*v*)	higher microbial diversity elevated *Firmicutes* (*Clostridiales*) the abundance of *Lachnospiraceae*, *Alistipes*, and *Odoribacter* significant differences among the three groups
Alcohol	Litwinowicz et al. 2020 [92]	systematic review	studies investigating intestinal microbiome alterations in individuals with alcohol use disorder (AUD) (*n* = 7)		depletion of *Akkermansia muciniphila,* and *Faecalibacterium prausnitzii* and an increase of *Enterobacteriaceae* in AUD higher abundance of *Proteobacteria* and lower of *Bacteroidetes*, lower abundance of several SCFAs-producing species
Alcohol	Addolorato et al. 2020 [93]	prospective, case-control, study	alcohol use disorder (AUD) patients (*n* = 36)	active drinkers vs. non-drinkers	decreased microbial alpha diversity in AUD reduction of *Akkermansia* and the increase of *Bacteroides* in AUD increased LPS and pro-inflammatory mediators increased in AUD
Coffee	Vitaglione et al. 2019 [94]	animal experimental study	C57BL/6J mice (*n* = 24)	standard diet, a high-fat diet (HFD), or an HFD plus decaffeinated coffee (HFD + COFFEE) for 12 weeks	HFD + COFFEE increased abundance of *Alcaligenaceae* in the feces
Coffee	Jaquet et al. 2009 [95]	interventional study	healthy adult volunteers (*n* = 16)	3 cups of coffee/day for 3 weeks	increase in the metabolic activity and/or numbers of the *Bifidobacterium* spp
Green tea	Seo et al. 2017 [96]	animal experimental study	C57BL/6J mice (*n* = 20)	fermented green tea 500 mg/kg/day for 8 weeks	reduced *Firmicutes/Bacteroidetes* ratio
Green tea	Liu et al. 2019 [97]	animal experimental study	C57BL/6J mice (*n* = 60)	HFD with 1% water extracts of green tea, oolong tea, and black tea	green tea: reduced plasma LPS, increased SCFAs production, decreased abundance of family *Rikenellaceae* and *Desulfovibrionaceae,* changed the abundance of OTU473 (*Alistipes*), OTU229 *(Rikenella*), OTU179 (*Ruminiclostridium*), and OTU264 (*Acetatifactor*)
Green tea	Yuan et al. 2018 [98]	interventional study	healthy volunteers (*n* = 12)	green tea liquid (GTL), (400 mL per day) 2 weeks	irreversibly, increase in *Firmicutes*: *Bacteroidetes* ratio, elevated SCFA-producing genera, reduced bacterial LPS
Pu-erh tea	Huang et al. 2019 [99]	an interventional, case-control study	human subjects and C57BL/6J mice	50 mg/kg/day for human subjects and 450 mg/kg/day for mice of Pu-erh tea	reduced *Lactobacillus, Bacillus, Streptococcus, and Lactococcus* genera in human subjects and mice
Salt	Bier et al. 2018 [100]	animal experimental study	Dahl salt-sensitive rats, 4 weeks old (*n* = 20)	normal diet (0.5% NaCl) vs. HSD (4% NaCl) for 8 weeks	HSD: an increased abundance of taxa from the Erwinia genus (*Christensenellaceae*, *Corynebacteriaceae*), the decrease in taxa from the *Anaerostipes* genus difference in fecal acetic acid, as propionic and isobutyric acid, but not in the butyric acid
Salt	Miranda PM et al. 2018 [101]	animal experimental study	six- to eight-week-old specific pathogen-free (SPF) male C57BL/6 mice	normal diet vs. HSD (4% NaCl) for 4 weeks	HSD: reduced *Lactobacillus* sp. and SCFAs production
Salt	Wang et al. 2017 [91]	animal experimental study	C57BL/6J mice	low- or high-salt diets (HSD) (0.25 vs. 3.15% NaCl) for 8 weeks	HSD increased *Firmicutes/Bacteroidetes* ratio, and the abundances of genera *Lachnospiraceae* and *Ruminococcus* (*p* < 0.05), but decreased the abundance of *Lactobacillus*
Salt	Wilck et al. 2017 [102]	animal experimental study	10-week-old, male C57BL6/J mice	normal salt (0.5% sodium) or high-salt diet (4% sodium + 1% in drinking water) ad libitum for 14 days	HSD created a distinct gut microbiome composition compared to the normal-salt diet (analysis of Jensen-Shannon divergence). HSD increased *Firmicutes:Bacteroidetes* ratio

Alcohol use disorder (AUD), green tea liquid (GTL), high-fat diet (HFD), high-salt diet (HSD), liposaccharides (LPS), short chain fatty acids (SCFAs), specific pathogen-free (SPF).

**Table 5 nutrients-12-01096-t005:** Influence of selected dietary patterns on GM.

Reference	Study Type	Population	Diet	Influence on Gut Microbiota
Genoni et al. 2019 [135]	cross-sectional comparative study	long-term (>1 year) (adult followers of a Paleolithic diet (PD) (*n* = 44) and controls (*n* = 47))	long-term Paleolithic diet (PD) vs. typical national recommendation (CD)	PD was associated with a higher abundance of TMA-producing *Hungatella*, higher TMAO vs. CD PD was inversely associated with the whole-grain intake
Barone et al. 2019 [136]	cross-sectional comparative study	healthy Italian subjects (*n* = 15) and urban Italian individuals (*n* = 143)	modern Paleolithic diet (MPD) vs. Mediterranean diet (MD) for one year	MPD was associated with the greater relative abundance of asaccharolytic bacteria (i.e., *Sutterella*, *Odoribacter*) and fat-and bile-tolerant bacteria (*Bilophila*)
Hansen et al. 2018 [139]	randomized, controlled, cross-over trial	healthy (non-celiac) adult Danish subjects (*n* = 60)	low-gluten diet (LGD) (2 g/d) vs. high-gluten diet (HGD) (18 g/d) for 8 weeks	LGD reduced *Bifidobacterium* spp. HGD decreased *Dorea longicatena* (and another species of *Dorea*), *Blautia wexlerae*, *Lachnospiraceae*, *Anaeostipes hadrus*, *Eubacterium hallii*
Bonder et al. 2016 [132]	observational study	healthy volunteers (*n* = 21)	gluten-free diet (GFD) for 4 weeks	GFD decreased *Veillonellaceae*, *Ruminococcus bromii*, and *Roseburia faecis*, *Victivallaceae*, *Clostridiaceae*, *ML615J-28*, *Slackia* GFD increased a *Coriobacteriaceae*
Özkul et al. 2019 [138]	pilot observational study	healthy adult men (*n* = 9)	Islamic fasting (IF) (17 h of fasting/day for 29 days)	IF increased abundance of *Akkermansia muciniphila* and *Bacteroides fragilis*
Xie et al. 2017 [140]	interventional study	children with drug-resistant epilepsy (*n* = 14)	ketogenic diet (KD) for 1 week	KD decreased the phylum *Proteobacteria* (*Cronobacter*) and increased the phylum *Bacteroidetes (Prevotella*, *Bifidobacterium*, *Bacteroides*)
Spinelli et al. 2018 [141]	interventional study	children with resistant epilepsy (*n* = 20)	ketogenic diet (KD) for 6 months	KD caused an overall decrease in the mean species diversity, KD increased *Bacteroides* and decreased *Firmicutes* and *Actinobacteria*

Casual diet (CD), islamic fasting (IF), gluten-free diet (GFD), high-gluten diet (HGD), ketogenic diet (KD), low-gluten diet (LGD), Mediterranean diet (MD), modern paleolithic diet (MPD), paleolithic diet (PD), trimethylamine N-oxide (TMAO).

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
