# Peer review of "You Are What You Eat—The Relationship between Diet, Microbiota, and Metabolic Disorders—A Review"

_nutrients, 2020, doi:10.3390/nu12041096_

Round 1

Reviewer 1 Report

The review explored the relationship between microbiome and diet, considering macronutrient and different dietary regimes and also the impact on metabolic disorders. It also discussed the dysbiosis development and its relation to metabolic disease. I found the review in a good structure. The summarized tables help the reader to gain good amount of information. The authors described in detail current knowledge on diet, going to the details of macronutrients, and microbiome and the dysbiosis of gut microbiome and different dietary patterns. The authors also tried to explore on the microbial metabolic production and their impact on host, however it was sufficed to SCFAs, TMAO, Bile acids and LP. Below is my comment to improve the work further.

Comments:

  • Abstract: Bacteriophages are virus, so I suggest to not separate them.
  • Introduction: Gut bacterial flora -> flora is collective bacteria, so I suggest use either of them.
  • Line 50: One of the key fermentative product of gut bacteria is Short chain fatty acids. I suggest including it with proper referencing. Also, recently there are publications on the amino acids metabolism and its importance in regulating host metabolism.

  • Line 69-> I don’t have objection to write like this but suggest making it like “unhealthy food, high alcohol consumption and bad nutritional habits.” There are sentences such so in the entire manuscript that would be better re-write them. Another example is line 84/85 change to maternal microbiome.

  • Line 155 missing “)” after CHO.

  • Line 195-> NAFLD hasn’t been introduce before.

  • References repeated twice in the table. I suggest o keep the last column and remove the first one.

  • There other typos that I suggest authors carefully go through the text one more time.

  • On section 3.2. The mechanism underlying gut microbiota-related metabolic disorders, SCFAs, bile acids and TMAO were reviewed. There is body of information on tryptophan derived microbial such as indole derivatives that has a crucial role on host physiology. For example: https://www.nature.com/articles/nature24661 and https://www.sciencedirect.com/science/article/pii/S2211124718304856                      I suggest the author to explore this further as diet also has effect on microbiome indole derivatives.

  • As the review explored the diet and microbial metabolism, I suggest adding few lines or section on state-of-art microbiome-based treatments such as personalized diet recommendation based on microbiome, metabolite supplement and etc. one example: https://www.sciencedirect.com/science/article/pii/S0092867415014816

  • Ending comes without a sharp conclusion and suffice to targeting microbiome for treatment. Again, the above comment could be considered to expand the conclusion with better future perspective.

Author Response

Dear Reviewer,

We are grateful to you for taking your time to read our paper and for your constructive comments. We have carefully reviewed the comments and have revised the manuscript accordingly. Our responses are given below in a point-by-point manner. Changes to the text are highlighted in the attached document using the "Track Changes" function in Microsoft Word, and the sections added to the manuscript are marked in red.

We hope the revised version is now suitable for publication and look forward to hearing from you in due course.

Sincerely,

Małgorzata Moszak

...................................................................................................................

Response to Reviewer Comments

Point 1. Bacteriophages are virus, so I suggest to not separate them.

Response 1. in line 10 and 34 (abstract and introduction) - the word "bacteriophages" has been removed

Point 2. Introduction: Gut bacterial flora -> flora is collective bacteria, so I suggest use either of them.

Response 2. in line 72 - the word “bacterial” has been removed

Point 3. Line 50: One of the key fermentative product of gut bacteria is Short chain fatty acids. I suggest including it with proper referencing. Also, recently there are publications on the amino acids metabolism and its importance in regulating host metabolism.

Response 3. The sentence has been rewritten: Previous studies showed that GM plays important roles in nutrient degradation and adsorption [3], short-chain fatty acids (SCFAs), amines, phenols/indoles, and sulfurous compounds production [8], vitamin B and K synthesis [9], the bioavailability of minerals, and the metabolism of bile acids [10].

Ref.: Oliphant, K.; Allen-Vercoe, E. Macronutrient metabolism by the human gut microbiome: major fermentation by-products and their impact on host health. Microbiome 2019, 7, 91.

Point 4. Line 69-> I don’t have objection to write like this but suggest making it like “unhealthy food, high alcohol consumption and bad nutritional habits.” There are sentences such so in the entire manuscript that would be better re-write them. Another example is line 84/85 change to maternal microbiome.

Response 4. Thank you very much for your comments regarding correctness and language diligence, which will allow us to improve our work in the future. At the same time, we'd like to assure you that our manuscript was edited with the help of Cactus Communications Korea Co. Ltd. (Editage).

in line 84/85 – the phrase “mother’s microbiomes” has been changed to “maternal microbiome”

Point 5. Line 155 missing “)” after CHO and Point 8. There other typos that I suggest authors carefully go through the text one more time.

Response 5 and 8. in line:154, 368, 513, 692 (and also in table 1, 2, 4, 5) some typographic errors in the manuscript have been corrected.

Point 6. Line 195-> NAFLD hasn’t been introduce before.

Response 6. in line 193/194 – the abbrev. “NAFLD” has been introduce

Point 7. References repeated twice in the table. I suggest o keep the last column and remove the first one.

Response 7. We decided to move the reference from the last column to the first column (and delete the last column):

  1. before

Reference

Study type

Population

Dietary sources of carbohydrate

Influence on gut microbiota

Ref.

Fehlbaum et al., 2018

in vitro study

screening platform (i-screen) inoculated with adult fecal microbiota

a different source of dietary fiber (DF):

FOS (chicory root), inulin (chicory root), alpha-GOS (peas), beta-GOS (lactose), XOS-C (corn cobs), XOS-S (sugar cane fiber) and β-glucan (oat flour)

β-glucan induced ↑ Prevotella and Roseburia and ↑ SCFA propionate production.

Inulin and GOS, XOS induced↑ Bifidobacteria

all DF had a prebiotic activity with β-glucan being dominant

[36]

  1. after

Reference

Study type

Population

Dietary sources of carbohydrate

Influence on gut microbiota

Fehlbaum et al., 2018

[36]

in vitro study

screening platform (i-screen) inoculated with adult fecal microbiota

a different source of dietary fiber (DF):

FOS (chicory root), inulin (chicory root), alpha-GOS (peas), beta-GOS (lactose), XOS-C (corn cobs), XOS-S (sugar cane fiber) and β-glucan (oat flour)

β-glucan induced ↑ Prevotella and Roseburia and ↑ SCFA propionate production.

Inulin and GOS, XOS induced↑ Bifidobacteria

all DF had a prebiotic activity with β-glucan being dominant

Point 9.

On section 3.2. The mechanism underlying gut microbiota-related metabolic disorders, SCFAs, bile acids and TMAO were reviewed. There is a body of information on tryptophan derived microbial such as indole derivatives that has a crucial role on host physiology. For example: https://www.nature.com/articles/nature24661 and https://www.sciencedirect.com/science/article/pii/S2211124718304856                      I suggest the author to explore this further as diet also has effect on microbiome indole derivatives.

Response 9. The following section has been added to the manuscript.

3.2.5 Tryptophan-derived metabolites

Tryptophan (Trp) is one of the essential aromatic amino acids supply from nutritional sources. About 4% to 6% of tryptophan is metabolized by GM into indican, indole or indole acid derivatives, skatole, and tryptamine [183]. The gut bacteria-derived indoles produced from tryptophan are a key modulator of host physiological and pathological pathways and hence may contribute to the cardiovascular, metabolic, and brain disorders [184]. The metabolism and biological effects of gut bacteria-derived indoles have been widely reviewed in several publications [184-186]. Indole propionic acid (IPA) is a metabolite produced by a small group of bacteria (especially C. sporogenes) from dietary tryptophan. Previous studies provided evidence that IPA plays an important role in intestinal barrier stabilization, antioxidant defense, and has neuroprotective effects [184]. The study conducted by Dodd et al. [187] in genobiotic mice inoculated with wild-type C. sporogenes showed that serum level of IPA was related to intestinal permeability and immune activation in a pregnane X receptor (PXR)-dependent fashion. The previous study comparing differences in gut microbiota-dependent metabolites between GF and conventionally raised mice contrary to mice fed a HFD or LFH revealed that indole-3-acetate (I3A) and tryptamine were depleted under a HFD. Both metabolites, especially I3A modulate inflammatory responses by decreased fatty-acid- and LPS-stimulated release of pro-inflammatory cytokines and chemokine [188].

Point 10. As the review explored the diet and microbial metabolism, I suggest adding few lines or section on state-of-art microbiome-based treatments such as personalized diet recommendation based on microbiome, metabolite supplement and etc. one example: https://www.sciencedirect.com/science/article/pii/S0092867415014816

Response 10. The following section has been added to the manuscript.

 3.2.6. The state-of-art: microbiome-based treatment The growing body of reports describing the diet-GM-host health relation has resulted in researches concerning the use of microbial remodeling strategies in the treatment of metabolic diseases. The recent systematic review on six RCTs study conducted by Tenorio-Jiménez et al. [189] evaluated that the probiotics intake in patients with MetS may improve some clinical parameters such as body mass index, blood pressure, glucose metabolism, lipid profile and inflammatory biomarkers but these beneficial effects seem to be clinically non-relevant [189]. However, it should be emphasized that the available data are still insufficient to recommend the supply of probiotics and/or prebiotics to all patients suffering from metabolic disorders explicitly. The rationalization of the use of probiotic-based strategies in the treatment of obesity and co-existing metabolic disorders requires the use of targeted types and strains of bacteria or/and their combinations as well as standardization of dose, time, and form of their supply. Identifying these elements seems to be critical to future research into the use of probiotics in the treatment of metabolic disorders. The personalized diet recommendation based on GM or using metabolite supplementation also may successfully modify various metabolic pathways and hence affect improvement in human health [190].

Point 11.

Ending comes without a sharp conclusion and suffice to targeting microbiome for treatment. Again, the above comment could be considered to expand the conclusion with better future perspective.

Response 11. The "conclusions" section has been reworded.

Long term dietary pattern appears to have a pivotal influence on GM composition and functionality and hence on host metabolism. Proper nutrition, defined as a calorie-balanced diet, with adequate fruit and vegetable intake, rich in dietary fiber, with healthy fats (MUFAs and PUFAs), and a predominance of plant-derived proteins seem to be the best way to promote GM diversity and activity. Previous studies in animals and humans have observed that human GM differ between obese and lean individuals or in individuals with metabolic disorders. The deregulation in gut microflora predisposes to impairment of host metabolism in many ways, including the bacterial influence on energy harvest from food, damage in intestinal epithelium tightness leading to metabolic endotoxemia, and alteration in the metabolism of carbohydrates, lipids, and bile acids. Therapies targeting GM dysbiosis may in the future be a promising tool in support of the treatment of patients with several diseases such as obesity, dyslipidemia, insulin resistance or type 2 diabetes. In-depth knowledge of connections between GM establishment, metabolism, and human health will allow to targeting microbiome reprogramming for the prevention and treatment of metabolic diseases in the future. The manipulation of microbiota-derived metabolites and their pathways can be helpful in exploring novel, individualized and efficient treatment modalities for various human disorders.

Reviewer 2 Report

Moszak et al. reviewed beautifully the connection between diet, microbiota, and metabolic disorders. There is not much to add into manuscript. It is well documented and concise. I have attached the pdf file where I underlined the minor corrections are needed.

Author Response

Dear Reviewer,

We are grateful to you for taking your time to read our paper and for your constructive comments. We have carefully reviewed the comments and have revised the manuscript accordingly. Our responses are given below in a point-by-point manner. Changes to the text are highlighted in the attached document using the "Track Changes" function in Microsoft Word.

We hope the revised version is now suitable for publication and look forward to hearing from you in due course.

Sincerely,

Małgorzata Moszak

Point 1.

line 60 - the word "two" has been changed to “2-5”, as the Reviewer suggested

line 83 – the word “three” has been changed to “2-5”, as the Reviewer suggested

Point 2.

line 108 – the word "Bifidobacterium" has been changed to italics  

Point 3 and Point 4.

line 154, 156 – some typographical errors have been corrected 

Point 5 and Point 8.

line 164, 166,242, 533,  - the word "Bifidobacterium" and “Lactobacillus” has been changed to italics 

Point 6.

line 198 – the Table 1 have been cited 

Point 7.

line 212 – some typographical errors have been corrected 
